computer modelling and simulation

cross-sectional data, predictive computational models, pseudo-longitudinal data, Langevin dynamics

**Author for correspondence:**
Rick Quax
e-mail: r.quax@uva.nl

# Inferring temporal dynamics from cross-sectional data using Langevin dynamics

Pritha Dutta[1], Rick Quax[2,3], Loes Crielaard[2,4,5], Luca Badiali[3] and Peter M. A. Sloot[2,6,7]

[1]Interdisciplinary Graduate Programme, Nanyang Technological University, Singapore
[2]Institute for Advanced Study, [3]Computational Science Lab, and [4]Amsterdam Public Health Research Institute, University of Amsterdam, Amsterdam, The Netherlands
[5]Department of Public and Occupational Health, Amsterdam UMC, Amsterdam, The Netherlands
[6]National Center for Cognitive Research, ITMO University, St Petersburg, The Russian Federation
[7]Complexity Science Hub, Vienna, Austria

RQ, 0000-0002-0299-0074

Cross-sectional studies are widely prevalent since they are more feasible to conduct compared with longitudinal studies. However, cross-sectional data lack the temporal information required to study the evolution of the underlying dynamics. This temporal information is essential to develop predictive computational models, which is the first step towards causal modelling. We propose a method for inferring computational models from cross-sectional data using Langevin dynamics. This method can be applied to any system where the datapoints are influenced by equal forces and are in (local) equilibrium. The inferred model will be valid for the time span during which this set of forces remains unchanged. The result is a set of stochastic differential equations that capture the temporal dynamics, by assuming that groups of datapoints are subject to the same free energy landscape and amount of noise. This is a 'baseline' method that initiates the development of computational models and can be iteratively enhanced through the inclusion of domain expert knowledge as demonstrated in our results. Our method shows significant predictive power when compared against two population-based longitudinal datasets. The proposed method can facilitate the use of cross-sectional datasets to obtain an initial estimate of the underlying dynamics of the respective systems.

## 1. Introduction

Longitudinal studies require a huge investment in terms of time, money and effort, depending on the system studied. For instance, biological experiment techniques such as sequencing-based assays destroy cells in order to measure certain

concentrations; population-based cohort studies in public health involve asking each of the participants to visit the hospital, measuring various physiological variables, and assessing psychological well-being through interviews and questionnaires. This leads to a relative abundance of cross-sectional datasets in these fields. However, the price to pay is that cross-sectional data lack the temporal information needed to study the evolution of the underlying dynamics. This hampers the development of models that can make predictions (predictive models) or even simulate the effects of interventions (causal models) in these fields. Therefore, in order to use the abundant cross-sectional data to study the dynamics of system behaviour it is important to design methods that aim to infer the temporal dynamics from these data.

There are several techniques in the literature for estimating pseudo-longitudinal data from cross-sectional data, which include employing distance metrics and graph theoretical operations [1–8]. They aim to construct realistic trajectories through the feature space by using techniques such as ordering of the data-points and selecting start-points and end-points based on known class labels. For instance, one way to order data-points is by assuming that the label 'healthy' precedes the label 'diseased'. Another way to order biological samples or RNA-seq data is by using their gene expression levels. Of course, these methods rely on the presence of suitable variables in the dataset, as well as the assumptions about how these labels induce an ordering. Our proposed method, on the other hand, infers the temporal dynamics from the distribution of the data-points, and hence, is not dependent on the ordering of the data-points or the presence of order-inducing variables.

In this work, we propose a method for inferring predictive computational models from cross-sectional data using Langevin dynamics [9]. The main step is to reconstruct a free energy landscape from the cross-sectional data by assuming that groups of data-points having similar features follow similar trajectories. A free energy landscape assigns an energy value to all possible states of the data-points in the system. Intuitively it can be considered analogous to an uneven hillside and the data-points in the landscape as balls rolling down the hillside. The balls will eventually come to rest in one or another local minimum (steady state) in the valleys. These valleys are the attractors in the free energy landscape. The sequence of states of a data-point over time traces a trajectory over this landscape. The trajectories of the data-points in our method are not only driven by the energy landscape's gradient but also by noise, which can be considered analogous to random small kicks applied continuously to all balls. In summary, our method estimates the free energy landscape based on the probability distribution of the data-points (which in turn is estimated from the data). More specifically, the free energy landscape is proportional to the logarithm of the inverse of the probability distribution of the data-points. In other words, the attractors in this estimated energy landscape will approximately correspond to the peaks in the probability distribution, and vice versa, peaks in the energy landscape will correspond to low-probability regions in the probability distribution.

The proposed method is based on the following assumptions. The first assumption is that the distribution depends only on the variable(s) of interest which are chosen to be *dynamic*. The second assumption is that nearby data-points have a statistical tendency to move in similar direction, i.e. downslope of a free energy landscape. Their exact trajectories at a particular time may nevertheless be very different, but this can only be due to the incidental noise which acts on all points at all times. The third assumption is that the distribution of the data-points is a sample from a distribution that is stable at the time of observation. For instance, if we have cross-sectional data concerning the BMIs of a group of individuals, we assume that the group of individuals is not currently subjected to any intervention that will change the distribution of BMIs within a short time span. That is, even though, the BMIs might change at the individual level, the distribution of the BMIs remains stationary at the population level. It is important to note that, in a group of data-points that exhibits a stationary distribution, no state is a permanent condition for any of the individual data-points, that is, the data-points undergo continual change. However, a force constrains the state space that can be explored by the data-points and this force is proportional to the gradient of the free energy landscape.

It is important to note that the appropriateness of these assumptions depends crucially on which variable(s) are considered 'dynamic', which variables are considered 'confounding', and which variables are considered 'independent'. The dynamic variables will form the dimensions of the energy landscape. For instance, if we have cross-sectional data concerning the variables BMI, physical activity (PA), diet and stress of a group of individuals, then we could (i) select BMI as our variable of interest to obtain a one-dimensional landscape over the BMI scale, or (ii) select both BMI and PA to make a two-dimensional landscape, and so on. This choice depends on which variables are expected to causally influence each other. To illustrate the consideration, suppose that a change in BMI is also

expected to change a person's PA (for instance, suppose a larger BMI would lead to lower PA and vice versa). In this case, it would be appropriate to create a two-dimensional landscape. Additionally, suppose that the landscape is such that, in order to go from a low-BMI state to a high-BMI state, one must first decrease the PA, and that this transition from high PA to low PA involves crossing a high-energy barrier (i.e. this transition has a low probability). In this case, it would be inaccurate to only consider BMI as 'dynamic', since this might lead the model to predict that an individual in the low-BMI state may readily progress towards the high-BMI state, whereas in reality the low-BMI state is relatively stable and the transition from the low-BMI state to the high-BMI state has low probability. However, considering PA as a second 'dynamic' variable may then match better with reality and make the transition of individuals from the low-BMI state to the high-BMI state less likely.

Confounding variables are those that are considered to have an effect on the energy landscape but that themselves are constant. Consider for instance the variable stress. Suppose that stress has an influence on which BMI values are attractor values; for instance, suppose that higher stress leads to a higher expected BMI value. Suppose further that stress itself is not influenced (appreciably) by BMI, but rather by long-term processes such as employment, socio-economic status and psychological traits, which act on timescales of years. On the other hand, our simulation may be designed to predict only for a few weeks or months. In this case, we may consider stress to be a confounding variable: stress has a causal effect on one or more of the dynamic variables, but is not causally influenced itself. This means that individuals with different stress levels may be following different corresponding landscapes and it is not possible to transition from one landscape to another.

Finally, independent variables are simply ignored from the model. That is, (i) they are assumed to have no effect on the dynamics or equivalently on the shape of the landscape(s), or (ii) their effect is already covered by (correlates strongly with) another variable that is already taken into account as dynamic or confounding, or (iii) their effect is assumed to be averaged out at the population level.

The temporal dynamics of the data-points are modelled using stochastic differential equations based on Langevin dynamics and the above-stated assumptions. The summed effect of all external factors and random incidents on the data-points is modelled using a noise term which causes random movement of the data-points within the landscape. By default, the added noise is statistically independent in all directions in our model. However, with the help of domain expert knowledge, this noise term can be defined by considering the dependencies on other variables and factors. It should be noted that since at the individual level the data-points exhibit random movement, the assumption that the data-points have a statistical tendency to move towards the stable attractor states is only valid at the population level and cannot be applied to the individual level.

This method is applicable to systems where the data-points are influenced by equal forces and are in (local) equilibrium. At the individual level, there may be continual change in the position of the data-points, but at the population level the distribution of these data-points remains stable. The inferred model will be valid for the time span during which this set of forces remains unchanged. This method would not be applicable to systems that are undergoing intervention. This is because the landscapes representing these systems will tend to change within short time spans. For example, a group of individuals for whom the current most probable BMI corresponds to overweight will follow a landscape where the attractor represents a BMI corresponding to overweight. However, if that group of individuals is undergoing an intervention that affects their weight-related behaviour, such as increased physical activity and a healthier diet, then the free energy landscape of that group will change with the attractor moving away from the BMI corresponding to overweight. In such a situation, our method will not be applicable since from a single cross-sectional dataset it will estimate only a single landscape. However, our method could be applied to this system when a past intervention has resulted in a (new) stable distribution of BMIs for the group of individuals, i.e. after a considerable time has passed from the start of the intervention.

The goal of our study is to formulate a method to infer a computational model that predicts the temporal evolution of the data-points. We would like to highlight here that our proposed method should only be viewed as a starting point for inferring temporal dynamics from cross-sectional data. Specifically, this method only presents a means to obtain an approximate idea of the underlying dynamics from the cross-sectional data without considering any other factors and dependencies. This method is not a causal inference technique in itself, which is a term reserved for automated techniques that identify causes of an effect by establishing that a cause-and-effect relationship exists purely based on data [10]. Our proposed method selects a stochastic differential equation model which best fits the landscape dynamics derived from the data. Even though a differential equation can always be interpreted as a causal model, since it specifies how one variable changes as function of

other variables, we would like to clarify that our predictive model does not by itself encode valid causal relationships. The selected model by our method (without further constraints) is therefore not causally interpretable. In other words, it can be used to predict the system behaviour, but not how this behaviour will change in response to interventions. To illustrate this point, consider a dataset of BMIs of a group of individuals. It may be reasonable to predict that on average this group of individuals will have a statistical tendency to move towards the most probable BMI within that group. However, it is an entirely different question what would happen to the BMIs of the individuals within this group after an intervention. Our method does not address the question of 'how' mechanistically the BMIs of the individuals within a group increase or decrease, only that on average they will statistically tend to increase or decrease.

Our model can be made causally interpretable by adding domain expert knowledge to our presented 'baseline' method in the form of (constraints on the) causal relationships between the different variables in the system. This should in turn increase the out-of-sample predictive power (generalizability). For example, in a previous work [11], we constructed an expert-informed causal model between individual body weight perception, individual weight-related behaviour, and group-level norms towards body weight. We incorporated domain expert knowledge to gather statements of causal and non-causal relationships to infer a computational model which can be causally interpreted. These statements of causal relations (e.g. 'physical activity directly affects weight loss') essentially constitute constraints on the functional forms of the differential equations that are permitted (the differential equation for 'weight loss' must at least include a variable 'physical activity'). The remaining uncertainties (parameter values, functional forms) were then estimated from a cross-sectional dataset by using assumptions. In addition, the selection of the variable(s) as 'dynamic', 'confounding' and 'independent' can also be made more accurate with the help of domain expert knowledge. Our proposed method can be considered as a starting point of this model-building process for systems that can be described as effectively following a free energy landscape that does not change within short time spans and is not under the influence of any external force: in its naive form it produces a predictive model, and the more expert-informed constraints on functional forms are added based on causal or non-causal statements, the more accurate the causal interpretation of the resulting model.

We present the theoretical foundation and a numerical illustration of the method which estimates the dynamics from a cross-sectional dataset and highlight the assumptions. We compare the estimates of the temporal dynamics obtained by our method against two population-based longitudinal datasets where the first time-point is used as the cross-sectional data and the subsequent time-points are used for comparison with the model predictions.

## 2. Methods

### 2.1. Langevin dynamics

We consider that each data-point, denoted as $x$, is vector-valued and can change over time. A data-point represents a set of attributes (such as BMI) of an individual. We assume that all data-points in a cross-sectional dataset of size $N$ have had sufficient time to mix and explore the state space by the time at which they are observed. Thus, we assume that even if there was a major perturbation, a system of data-points has converged to a stable distribution at the time of our observation. This implies that all the data-points follow a probability distribution, $p(x)$, which is stationary. Furthermore, this implies that $N$ is large enough to effectively estimate $p(x)$ from the dataset.

In general, it is not possible to derive the temporal evolution, $dx/dt$, from the stationary probability distribution $p(x)$. This is because there are multiple $dx/dt$ which can lead to the same stationary distribution. For example, a data-point rotating clockwise and another data-point rotating anti-clockwise both have the same circular stationary distribution, but their $dx/dt$ are different. We solve this problem by assuming that each data-point tends to follow the same free energy landscape, $F(x)$, in a 'downslope' manner, i.e. moves in time in the direction of $-dF/dx$. $F(x)$ effectively assigns an energy value to each possible vector $x$. The assumption is that systems have a tendency to minimize their energy, although random fluctuations may sometimes have the opposite effect. The minimum energy states correspond to the attractors in $F(x)$.

The energy landscape, $F(x)$, can be derived from $p(x)$ through the Boltzmann distribution [12],

$$\beta F(x) = -\log p(x). \tag{2.1}$$

Here, the constant $\beta$ is interpreted as the inverse of temperature, or noise level: the lower the value of $\beta$, the larger is the effect of random fluctuations on $x$, and thus the lower the influence of $F(x)$ on the data-points. The assumption behind this relation is that the data-points are in equilibrium, which is already met by the assumption of $p(x)$ having reached stationarity.

In addition to this deterministic tendency given by $F(x)$, there is a random movement (noise) which is irrespective of $F(x)$ and is uncorrelated over time, defined by a Wiener process $W(t)$ [13]. Thus, the displacement of each data-point consists of the sum of a deterministic component and a stochastic component which leads to the following overdamped Langevin dynamics equation [9]:

$$\mathrm{d}x = -\beta \frac{\mathrm{d}F(x)}{\mathrm{d}x}\,\mathrm{d}t + \sigma\,\mathrm{d}W(t). \tag{2.2}$$

Here, $\beta$ controls the relative strength of the deterministic force and $\sigma$ controls the relative strength of the noisy movement. When $\beta \to 0$, the deterministic component becomes negligible, and thus the data-points diffuse randomly over the state space in all directions. When $\sigma \to 0$, the stochastic component becomes negligible, and all data-points converge to one or more local minima of $F(x)$, after which no further change occurs. Clearly, a balance is needed between these two opposing processes, which will control the degree of clustering and the amount of variance observed in the predicted distribution, $\hat{p}_{\beta,\sigma}(x)$, and match with the $p(x)$ from the data.

For determining the values of $\beta$ and $\sigma$, it is important to realize that only their ratio changes the stationary data distribution. Thus, if we fix the timescale of the deterministic component by setting $\beta = 1$ without loss of generality, then the remaining free parameter $\sigma$ controls the ratio of the random and deterministic forces. This timescale can be fixed because if we multiply both parameters in equation (2.2) with a constant $\tau$, the velocity $\mathrm{d}x/\mathrm{d}t$ is also multiplied by $\tau$. This means that $x$ changes $\tau$ times faster. In principle, it is impossible to derive how fast $x$ changes per unit time from the data available at a single time-point. We can, however, still predict the directions of the data-points, which is our main interest.

## 2.2. Numerical algorithm

To explain the numerical algorithm we will generate a set of data-points moving over a known free energy landscape and show how the algorithm recovers the dynamics. Consider a true underlying free energy landscape, $F(x) = -ax^2 + bx^4$, from Landau's second order phase transition formalism, which contains two attractor states as long as $a > 0$, $b > 0$. We set $a = 3$ and $b = 1$, where $a$ controls the height of the energy barrier separating the two attractors. Assuming the Boltzmann distribution (equation (2.1)), we generate a dataset consisting of 5000 i.i.d. data-points from the probability density function, $p(x) = e^{ax^2 - bx^4}/Z$, where $Z$ is the normalizing constant. $p(x)$ is shown in figure 1a as a solid red line. The sampling is done through the inverse transform sampling method [14].

We will now treat this dataset as given and 'forget' the $F(x)$ used to generate it. The true $p(x)$ always has an exponential form due to the assumption of a Boltzmann distribution. Therefore, a Gaussian kernel density estimation algorithm is used to estimate the data distribution. A Gaussian kernel density estimate is defined as $\hat{p}(x) = (nh\sqrt{2\pi})^{-1}\sum_i^n e^{(x-x_i)^2/2h^2}$, where $x_i$ are the data-points and the parameter $h$ is determined using Silverman's optimization procedure [15]. We would like to highlight here that even though the parameters $h$ and $\sigma$ represent similar concepts (both are standard deviation controlling parameters), they implement different components. The parameter $h$ controls the standard deviation of the Gaussian kernel density estimate obtained from the data, whereas $\sigma$ controls the ratio of the random and deterministic forces in equation (2.2). The estimated $\hat{p}(x)$ from the dataset is shown in figure 1a as a dashed black line. This estimated $\hat{p}(x)$ is then used to estimate the free energy landscape as $\hat{F}(x) = -\log \hat{p}(x)$. By fixing the timescale of the deterministic dynamics through $\beta = 1$, the deterministic term, $\mathrm{d}F/\mathrm{d}x$, of the Langevin dynamics (equation (2.2)) is given as

$$\frac{\mathrm{d}F}{\mathrm{d}x} = -\sum_{i=1}^{n} \frac{(x_i\,e^{-((x-x_i)^2)/2h^2}/h^2 - (x\,e^{-((x-x_i)^2)/2h^2}/h^2))}{e^{-((x-x_i)^2)/2h^2}}. \tag{2.3}$$

The stochastic part is then fully determined by an optimal choice for $\sigma$. The optimal $\sigma$ is determined by using the Hellinger distance, $H(\hat{p}_{\beta=1,\sigma}(x), p(x))$, as the cost function. The Hellinger distance is defined as, $H(p,q) = \frac{1}{2}\int_x (\sqrt{\mathrm{d}p(x)} - \sqrt{\mathrm{d}q(x)})^2\,\mathrm{d}x$. To evaluate the goodness of fit for each choice of $\sigma$ during this optimization process, we need to estimate the stationary probability density $\hat{p}_{\beta=1,\sigma}(x)$ numerically. Since integration techniques for stochastic differential equations are computationally intensive, we choose to discretize the domain of the data, $x$, into $f$ discrete points with distance $\Delta x$, which is calculated as

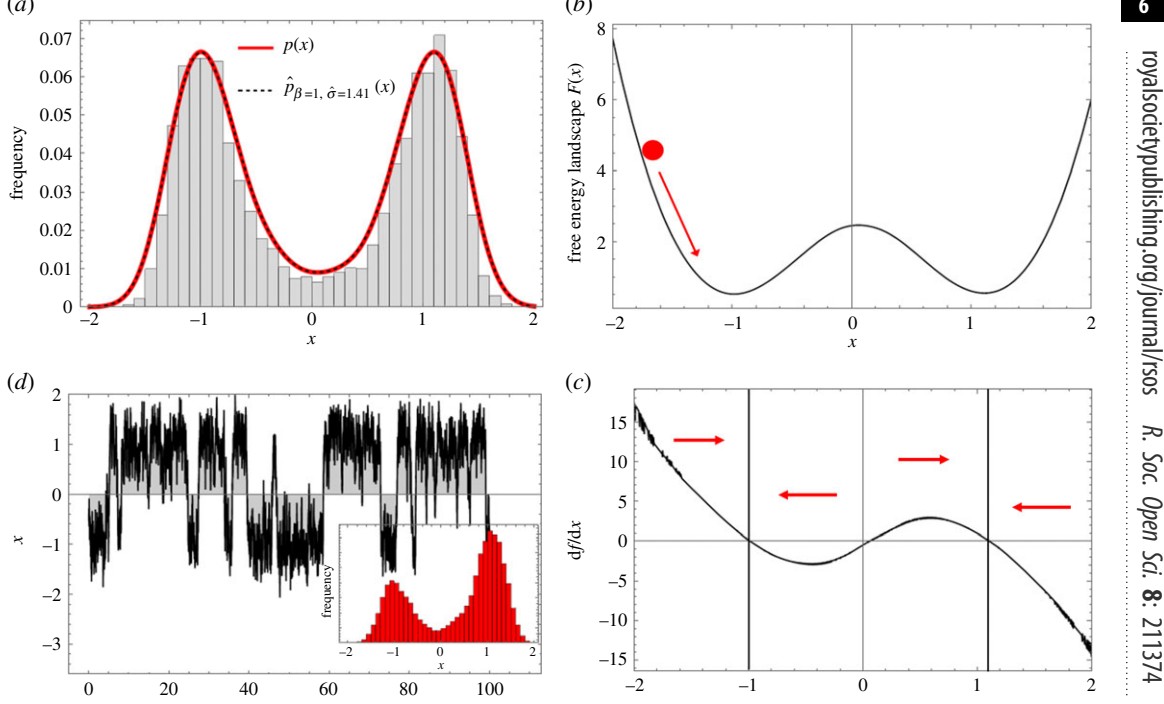

**Figure 1.** The proposed procedure to infer a computational model from a cross-sectional dataset. (*a*) The energy landscape $F(x) = -ax^2 + bx^4$ is used to generate a surrogate dataset, using the true probability density (solid red line). The dashed black line is the estimated probability density. (*b*) From the estimated probability density and the Boltzmann distribution we estimate the free energy landscape $F(x)$. The red ball represents a data-point moving downslope towards the attractor as shown by the red arrow. (*c*) The force field which is the derivative of the free energy landscape leads to the deterministic dynamics of the Langevin equation. The red arrows show the direction in which a data-point in any of the four partitions will be forced to move by the force field. The regions around 1 and $-1$ correspond to the two attractors. (*d*) The resulting predicted dynamics of a single data-point and the occasional transition from one attractor to the other, across the tipping point at $x = 0$. The inset figure is the histogram of the values of this data-point across time. The histogram almost captures the shape of the histogram of the cross-sectional data shown in (*a*) and is slightly different because we have taken a short time range to improve visibility of the predicted dynamics.

$\Delta x = \sqrt{\Delta t}/f$. Here, $\sqrt{\Delta t}$ is the expected displacement of a random diffusion process after $\Delta t$ time. We find that $f = 10$ produces accurate results at low memory cost. $x_{\min}$ and $x_{\max}$ are calculated from the data by taking the floor of the minimum value and ceiling of the maximum value from the standardized data respectively. In this case, we get $x_{\max} = -x_{\min} = 3$ from the generated dataset. We obtain 601 discrete values for $x$ and a sparse, approximately band-diagonal transition matrix of dimension $601 \times 601$, denoted by $M$. To calculate the transition probabilities in $M$, we first determine the possible positions that a data-point, starting at position $x$, can reach after $\Delta t$ time-steps. The displacement due to the deterministic force is given by $x + (-(\mathrm{d}F/\mathrm{d}x)\Delta t)$. Taking this value as the mean, we determine from the 601 discrete values of $x$ the values ($y$) that fall within four standard deviations of this mean, i.e. within $\pm 4\sigma\sqrt{\Delta t}$ of the mean. We then calculate the probabilities of displacement to these $y$ values due to random diffusion, by assuming a normal distribution with mean as $x + (-(\mathrm{d}F/\mathrm{d}x)\Delta t)$ and standard deviation as $\sigma\sqrt{\Delta t}$. Thus, $M_{x \xrightarrow{} y} = \mathcal{P}(y \mid \mathcal{N}(x - (\mathrm{d}F/\mathrm{d}x)\ \Delta t, \sigma\sqrt{\Delta t}))$, where $\mathcal{P}(.)$ denotes a probability density function.

For discrete Markov processes, the stationary distribution vector $\pi$ can be found directly by solving the set of linear equations $\pi = \pi M$, which is computationally an efficient operation since it reduces to finding the (left) eigenvector of $M$, having an eigenvalue of 1. Before performing this operation, we normalize the rows of $M$. The initial distribution vector, $\pi_0$, is calculated from the data as, $\pi_0(x) = \int_{x-\Delta x/2}^{x+\Delta x/2} p(x)$. Finally, we compute the Hellinger distance, $H(\hat{p}_{\beta=1,\sigma}(x), p(x))$, for which $\pi$ is converted to a continuous function, $\hat{p}_{\beta=1,\sigma}(x)$, using third-order spline interpolation. If changing $\sigma$ no longer reduces the Hellinger distance, the procedure terminates and the Langevin dynamics (equation (2.2)) is completely specified. In figure 1*d*, the resulting predicted dynamics of a data-point is shown, where the optimal $\hat{\sigma} \approx 1.41$.

This free energy landscape can be manipulated to favour a particular attractor or shift the attractor. Here, we show that the landscape can be changed by an intervention, and if it is changed then what

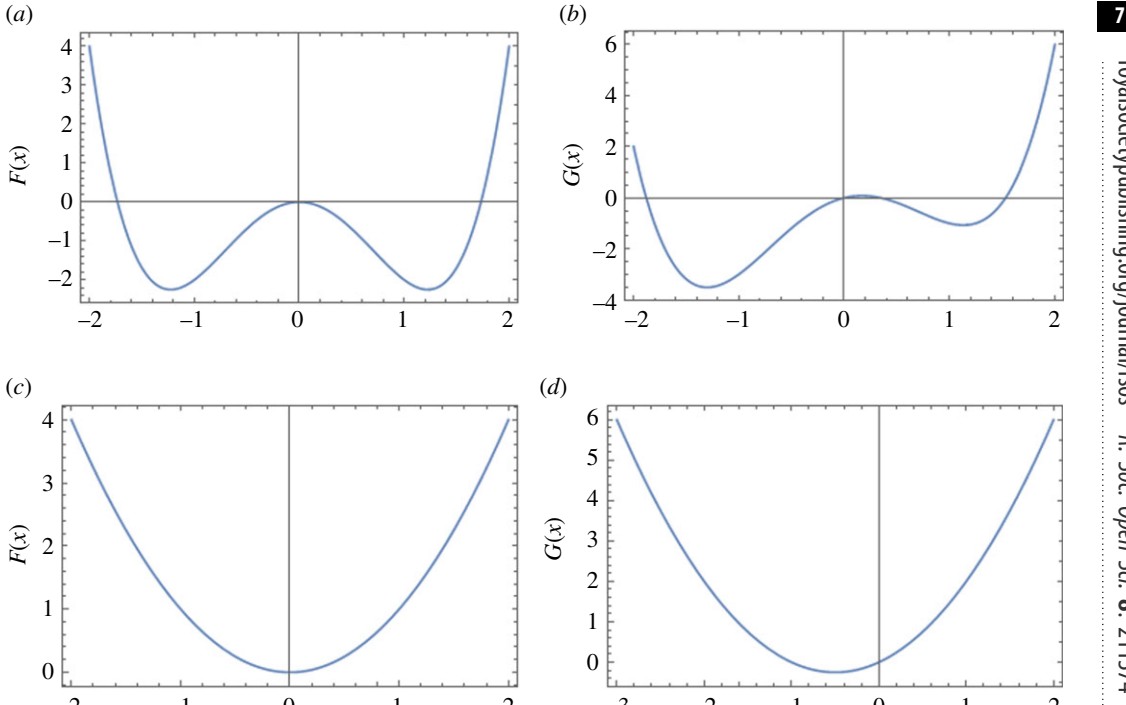

**Figure 2.** (a) Two-attractor pre-intervention landscape, (b) two-attractor post-intervention landscape, (c) single-attractor pre-intervention landscape, (d) single-attractor post-intervention landscape.

possible comparisons can be performed using the pre-intervention and post-intervention landscapes. We would like to point out that we do not know *how* an intervention should be implemented. This should be viewed as only a theoretical test where a term (analogous to an intervention) is added to the free energy function, and then potential comparison measures between the pre-intervention and post-intervention cases are presented.

Suppose, $F(x)$ denotes the pre-intervention free energy landscape and $G(x) = F(x) + cx$ represents the post-intervention free energy landscape. Here, $cx$ represents the intervention and $c$ controls the strength of the effect of the intervention. Let, $x_1(t)$ be a stochastic process with respect to the landscape $F(x)$ defined as

$$\mathrm{d}x_1 = -\frac{\mathrm{d}F}{\mathrm{d}x_1}\,\mathrm{d}t + \sigma\,\mathrm{d}W(t). \tag{2.4}$$

Let, $x_2(t)$ be another stochastic process with respect to the landscape $G(x)$ defined as

$$\mathrm{d}x_2 = -\frac{\mathrm{d}G}{\mathrm{d}x_2}\,\mathrm{d}t + \sigma\,\mathrm{d}W(t). \tag{2.5}$$

For a two-attractor landscape, suppose the pre-intervention free energy landscape is defined as $F(x) = -ax^2 + bx^4$. Adding a term $cx$ makes the left attractor preferred. The pre-intervention and post-intervention landscapes are shown in figure 2a,b. The average rate of transition from the attractor at $x_+$ to the attractor at $x_-$, with $x_0$ as the local maximum (tipping point) can be approximated using Kramer's formula for energy barrier crossing as

$$r(+ \rightarrow -) = \frac{1}{2\pi}\sqrt{F''(x_+)|F''(x_0)|}\,\mathrm{e}^{-2((F(x_0)-F(x_+))/\sigma^2)}, \tag{2.6}$$

under the condition that the exponent $|F(x_0) - F(x_+)|/\sigma^2/2 >> 1$. Similarly, the average rate of transition in the opposite direction, $r(- \rightarrow +)$ can be obtained by

$$r(- \rightarrow +) = \frac{1}{2\pi}\sqrt{F''(x_-)|F''(x_0)|}\,\mathrm{e}^{-2((F(x_0)-F(x_-))/\sigma^2)}, \tag{2.7}$$

under the condition that the exponent $|F(x_0) - F(x_-)|/\sigma^2/2 >> 1$.

Now, for the pre-intervention landscape, $F(x)$, in our example, the average transition rates in both direction will be equal since the landscape is symmetric. Using $a = 3$, $b = 1$, we get $r(+ \rightarrow -) = r(- \rightarrow +) = 12.9867$. For the post-intervention landscape, $G(x)$, the average rate of transition from the right attractor at $x_+$ to the left attractor at $x_-$ will be higher than that in the opposite direction. Using $a=3$, $b=1$ and $c=1$, we get $r(+ \rightarrow -) = 0.448$ and $r(- \rightarrow +) = 0.031$.

For a single-attractor landscape, suppose the pre-intervention free energy landscape is defined as $F(x) = ax^2$. Adding a term $cx$ shifts the attractor to the left. The pre-intervention and post-intervention landscapes are shown in figure 2c,d. In this case, we can compare the first moments (means) of the pre-intervention and post-intervention stochastic processes $x_1(t)$ and $x_2(t)$ respectively. Let $\mu_1$ and $\mu_2$ be the means of the stochastic processes $x_1(t)$ and $x_2(t)$, respectively, and they are defined as

$$\mu_1 = e^{-2at}x_{10} \tag{2.8}$$

and

$$\mu_2 = \frac{e^{-2at}(c + 2ax_{20}) - c}{2a}, \tag{2.9}$$

where, $x_{10}$ and $x_{20}$ are the initial states of $x_1(t)$ and $x_2(t)$, respectively.

## 2.3. Comparison with longitudinal dataset

We compare the estimates of the temporal dynamics obtained by our method against longitudinal datasets where the first time-point is used as the cross-sectional data and the subsequent time-points are used for comparison with the model predictions. For this purpose, we estimate the displacement of each individual which is influenced by a force towards the attractor and a random movement. Thus, this estimated displacement is not deterministic, but is random and hence represented as a normal distribution, whose mean is the gradient of the free energy landscape and standard deviation is the noise parameter. Mathematically, the displacement of an individual $i$ for a $\Delta t$ time-step is represented by a normal distribution with the displacement due to the deterministic force, $(-dF/dx)\Delta t$ as the mean and $\sigma\sqrt{\Delta t}$ as the standard deviation. Using this normal distribution, we determine for each individual $i$ the probability of positive displacement and negative displacement, denoted as $P_{PD}^i$ and $P_{ND}^i$, respectively and defined as:

$$P_{PD}^i = P\left(x > 0, x \sim \mathcal{N}\left(-\frac{dF}{dx}\Delta t, \sigma\sqrt{\Delta t}\right)\right) \tag{2.10}$$

and

$$P_{ND}^i = P\left(x < 0, x \sim \mathcal{N}\left(-\frac{dF}{dx}\Delta t, \sigma\sqrt{\Delta t}\right)\right). \tag{2.11}$$

Here, we quantify the prediction accuracy of our model for individual $i$, which we denote as $A^i$. If the observed displacement from the data, $D_{data}$, for individual $i$ is positive, we assign $P_{PD}^i$ as the prediction accuracy of our model, and if the observed displacement is negative, we assign $P_{ND}^i$ as the prediction accuracy of our model for individual $i$. The average prediction accuracy of the model, denoted as $A^{average}$, is then determined as the average of the prediction accuracy over all individuals. Mathematically, $A^i$ and $A^{average}$ are defined as

$$A^i = \begin{cases} P_{PD}^i, & \text{if } D_{data} \in \mathbb{R}_{>0}, \\ P_{ND}^i, & \text{if } D_{data} \in \mathbb{R}_{<0} \end{cases} \tag{2.12}$$

and

$$A^{average} = \langle A^i \rangle_i. \tag{2.13}$$

We also determine the maximum prediction accuracy of our model for individual $i$, $A_{max}^i$, and the average maximum prediction accuracy of our model over all individuals, $A_{max}^{average}$. These two quantities are defined as

$$A_{max}^i = \max(P_{PD}^i, P_{ND}^i) \tag{2.14}$$

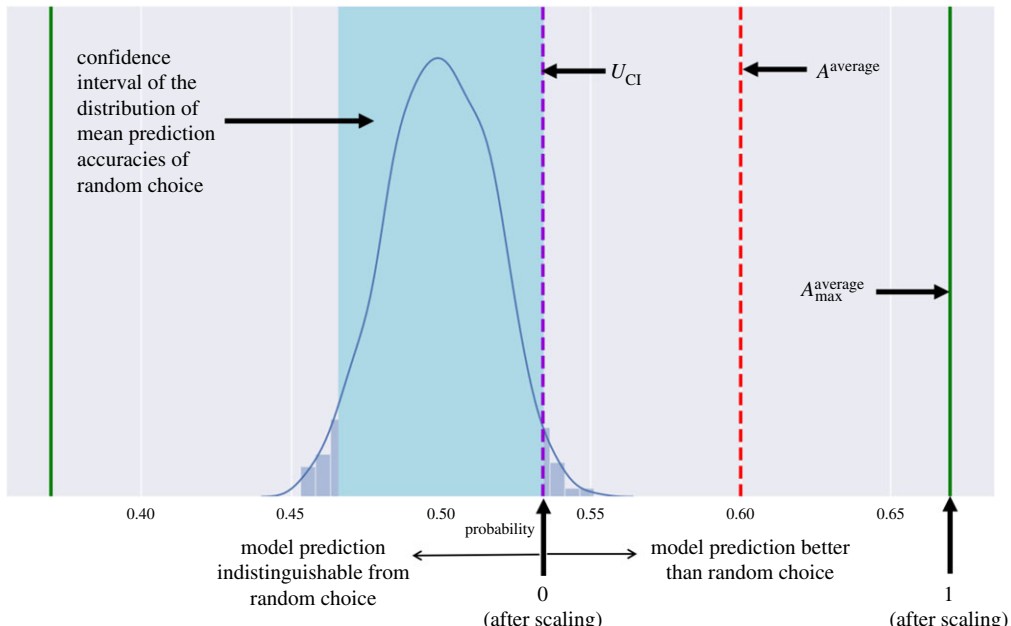

**Figure 3.** Comparison of the prediction accuracy of our model to random choice under the null hypothesis of random displacement. The blue shaded area is the confidence interval of the distribution of 1000 mean prediction accuracies of random choice. The violet dashed line denotes the upper limit ($U_{CI}$) of this confidence interval (equation (2.16)). The red dashed line denotes average prediction accuracy ($A^{average}$) (equation (2.13)) and the green solid line denotes the average maximum prediction accuracy ($A_{max}^{average}$) (equation (2.15)) (refer to section 2.3: Comparison with longitudinal dataset for details).

and

$$A_{max}^{average} = \langle A_{max}^i \rangle_i. \tag{2.15}$$

To determine the optimal value of $\Delta t$, we employ the metric of Euclidean distance, by varying its value from 1 to 100 with increments of 1, and selecting the value that gives the minimum Euclidean distance between the displacement due to the deterministic force ($-(dF/dx)\Delta t$) predicted by our model and the observed displacement from data.

In addition to the above tests, we compare the prediction accuracy of our model to random choice under the null hypothesis of random displacement. We randomly choose between $P_{PD}^i$ and $P_{ND}^i$ for each individual $i$ and then take the average over all individuals to obtain the prediction accuracy by random choice. We repeat this random choice step 1000 times to obtain a distribution of mean prediction accuracies of random choice and then determine the 95% confidence interval of this distribution. The upper limit of this confidence interval is defined as

$$U_{CI} = \overline{X} + 1.96 \frac{s}{\sqrt{1000}}, \tag{2.16}$$

where, $\overline{X}$ is the mean of the distribution of 1000 mean prediction accuracies of random choice and $s$ is the standard deviation of this distribution. If the average prediction accuracy of our model is greater than the upper limit ($U_{CI}$) of this confidence interval, we can say that our model prediction is significantly better than the prediction obtained by random choice. We also scale the average prediction accuracy ($A^{average}$) to better compare with $U_{CI}$ and the average maximum prediction accuracy ($A_{max}^{average}$) as

$$A_{scaled}^{average} = \frac{A^{average} - U_{CI}}{A_{max}^{average} - U_{CI}}. \tag{2.17}$$

After the above scaling, $U_{CI}$ will correspond to zero and $A_{max}^{average}$ will correspond to 1. If $A_{scaled}^{average}$ is above zero, we say that our model prediction is better than random choice; if it is below zero, then our model prediction is indistinguishable from random choice. This comparison process with random choice is explained in figure 3. Thus, we compare our model predictions against data as well as test whether the prediction accuracy of our model is significantly better than that by random choice.

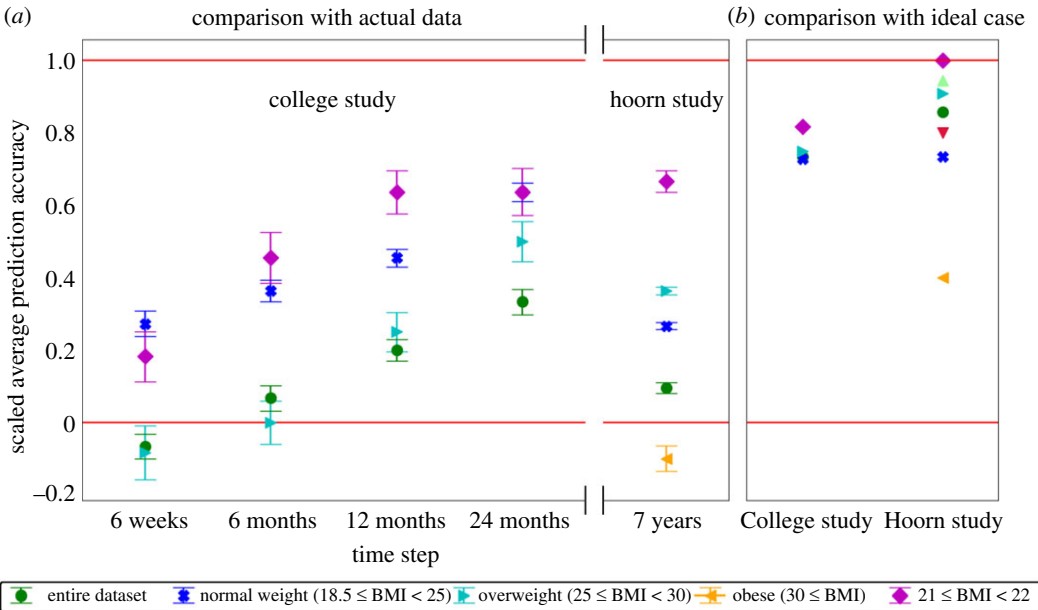

**Figure 4.** (a) The scaled average prediction accuracies, $A_{scaled}^{average}$ (equation (2.17)) corresponding to the four time-points: six weeks, six months, 12 months and 24 months of the College study dataset, and the single time-point of 7 years of the Hoorn study dataset. The vertical bars at each point represent the error bar for the respective model prediction accuracy calculated from 1000 bootstrap samples. The solid red line at 0 corresponds to the upper limit of the 95% confidence interval of the distribution of mean prediction accuracies of random choice, $U_{CI}$ (equation (2.16)), and the solid red line at 1 corresponds to the average maximum model prediction accuracy, $A_{max}^{average}$ (equation (2.15)). Model prediction accuracies above the line at 0 are significantly better than the prediction accuracy of random choice. (b) The 'maximally achievable' scaled average prediction accuracy of our model obtained by comparing against the theoretical test that the statistical tendency to move towards the attractor which is displayed at the population level was also displayed at the individual level, that is, if all individuals have a tendency to move towards the attractor (refer to section 3: Results).

## 2.4. Data

We compare our method against two population-based longitudinal datasets: the College study dataset [16] and the Hoorn study dataset [17–19], where we consider the first time-point as the cross-sectional data and the subsequent time-points are compared with the model prediction.

The College study dataset [16] is a longitudinal dataset with weights and heights measured over five time-points: baseline, six weeks, six months, 12 months, and 24 months. This study was conducted with 294 female first-year students (age: $18.24 \pm 0.44$ years; all values are expressed as mean $\pm$ s.d. unless otherwise specified) recruited from two universities in Philadelphia. We preprocessed this dataset to include only those participants who had their weight and height measured for all the five time-points and obtained 162 participants. We calculated the BMIs (in $kg\,m^{-2}$) of these 162 participants from their weight and height and their baseline BMI distribution is $23.59 \pm 2.69\ kg\,m^{-2}$.

The Hoorn study dataset [17–19] is a longitudinal dataset with BMIs measured over two time-points: baseline, and 7 years. The baseline study [17,19] was conducted in 2006–2007 in the Dutch city of Hoorn and included 2807 participants. The follow-up study [18,19] was conducted in 2013–2015 and included 1734 participants out of the 2807 participants in the baseline study. We preprocessed this dataset to include only those participants who had their BMIs measured for both time-points and obtained 1727 participants with age $53.62 \pm 6.53$ years and baseline BMI distribution $26.11 \pm 3.88\ kg\,m^{-2}$.

## 3. Results

We present a baseline method for inferring predictive computational models from cross-sectional data based on Langevin dynamics. We compare our model predictions against two longitudinal datasets: the College study dataset [16] and the Hoorn study dataset [17–19], where we consider the first time-point as the cross-sectional data and the subsequent time-points are compared with the model predictions. The College study dataset contains BMIs over five time-points: baseline, six weeks, six months, 12 months, and 24 months. The Hoorn study dataset contains BMIs over two time-points: baseline, and 7 years. Figure 4 shows our model prediction accuracies ($A_{scaled}^{average}$ (equation (2.17))) using

these two datasets. Our model shows significant predictive power (green circles in figure 4a) compared with the prediction of a random choice algorithm (solid red line 0 in figure 4). Additionally, our model prediction accuracy improves further when we incorporate domain expert knowledge to our model, as shown by the blue crosses, cyan left triangles, and magenta diamonds in figure 4a.

To test if the predictive power of our model is enhanced with the incorporation of domain expert knowledge, we apply the empirical observation from epidemiology that individuals from different BMI categories follow different landscapes [20,21]. In accordance, we cluster the datasets based on the standard BMI categories: underweight (BMI < 18.5), normal weight ($18.5 \leq$ BMI < 25), overweight ($25 \leq$ BMI < 30), and obese ($30 \leq$ BMI) [22]. We respectively obtain clusters of sizes 0, 119, 39 and 4 from the College study dataset, and 12, 739, 729 and 247 from the Hoorn study dataset. Since the underweight and obese clusters from the College study dataset have only 0 and 4 individuals, respectively, and the underweight cluster from the Hoorn study dataset has only 12 individuals, we disregard those clusters. We observe that if we consider separate attractor landscapes for these different clusters, the model prediction accuracy increases significantly (figure 4a). The attractor in each cluster approximately corresponds to the BMI that is most prevalent relative to the group of individuals in that cluster. From the above results (figure 4a), we can conclude that the incorporation of domain expert knowledge to our baseline method further enhances the predictive power of our model. Next, we do a theoretical test by further narrowing the BMI range to see if clustering individuals having almost exactly the same BMIs further improves the prediction accuracy. Accordingly, we select all individuals having BMIs in the narrow range of $21 \leq$ BMI < 22, and obtain 29 individuals from the College study dataset, and 96 individuals from the Hoorn study dataset. We observe that the model prediction accuracy increases further (figure 4a).

Next, we do a theoretical test to see how our method performs if the statistical tendency to move towards the attractor that is displayed at the population level was also displayed at the individual level, that is, if all individuals have a tendency to move towards the attractor. In that case, individuals with a BMI that is greater than the attractor BMI would decrease their BMI and vice versa. Assuming this, we determine the 'maximally achievable' prediction accuracy of our model. As observed from figure 4b, these 'maximally achievable' model prediction accuracies are significantly higher than the actual prediction accuracies using the real data.

We do another theoretical test by analysing the data to see if individuals having the same BMI have displacements in the same direction. We select 15 BMI bins based on the data (the bins are shown as x-axis labels in figure 5). We place an individual in BMI bin x if the individual's BMI falls in the range of $x \leq$ BMI < x + 1. For example, we place an individual in BMI bin 25 if the individual's BMI falls in the range of $25 \leq$ BMI < 26. Then, we calculate the displacements from baseline to six weeks, six months, 12 months and 24 months. Figure 5 shows the relative number of individuals having positive and negative displacements in each BMI bin, which is calculated as (number of positive displacements − number of negative displacements)/(number of positive displacements + number of negative displacements). If all individuals in a particular BMI bin have displacements in the same direction, then this relative number will be 1 or −1 as shown by the red solid lines in figure 5. If individuals in a particular BMI bin have mixed displacement directions then this relative number will be between 1 and −1. With this figure we want to show that individual behaviour is inherently random and that individuals having almost the same BMI may not have displacements in the same direction. That is, the weights of two individuals having the same BMI of 28 may not decrease in both cases: in one case it may increase, whereas in the other it may decrease. If all individuals in a particular BMI bin had displacements in the same direction, then the relative number of individuals having positive and negative displacements in each BMI bin would be exactly 1 or −1. However, this is not what we observe from figure 5. It should be noted that this analysis is based on the assumption that the distribution does not depend on other variables, which in reality is probably not true. And it is because of these other variables that the relative number of individuals having positive and negative displacements in each BMI bin is not exactly 1 or −1. However, the purpose of this analysis is to show that, even without considering the other variables and factors, our method is already able to provide a good starting estimate of the underlying dynamics from the cross-sectional data. This estimate can be further enhanced by adding domain expert knowledge in the form of (constraints on the) causal relationships between the different variables in the system.

Next, we test the performance of our method when the cross-sectional dataset is small. For this purpose, we use the free energy landscape, $F(x) = -ax^2 + bx^4$, from Landau's second order phase transition formalism to generate data-points as shown in §2.2: Numerical algorithm. We generate a large dataset consisting of 5000 data-points and 1000 small datasets consisting of 40 data-points. Figure 6 shows the estimated probability densities from the large dataset (in red) and the 1000 small

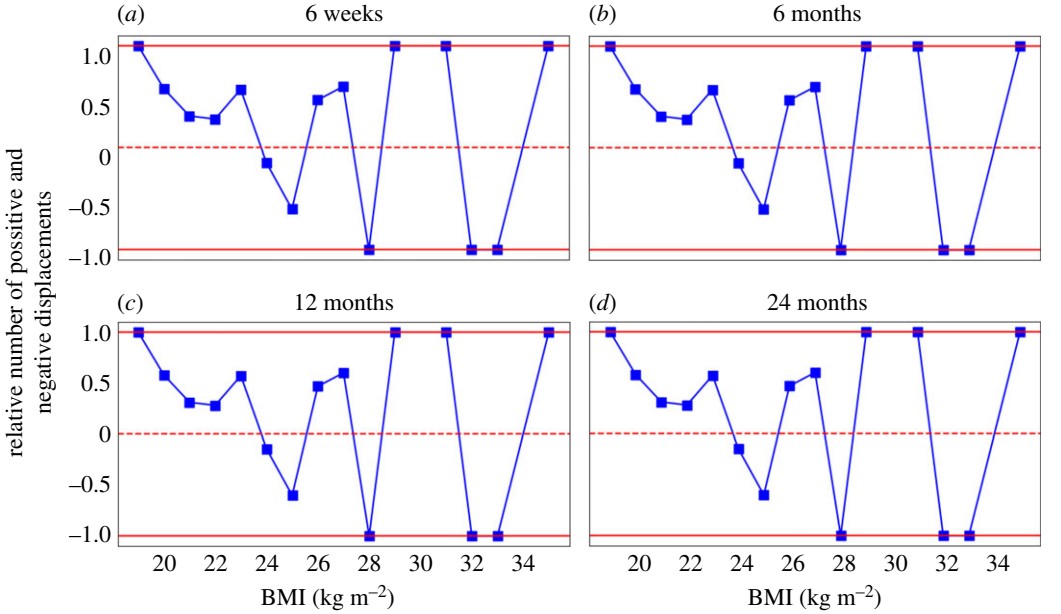

**Figure 5.** The relative number of positive and negative displacements in each BMI bin at (*a*) six weeks, (*b*) six months, (*c*) 12 months and (*d*) 24 months obtained from the College study dataset. The relative number of positive and negative displacements in each BMI bin is calculated as (number of positive displacements − number of negative displacements)/ (number of positive displacements + number of negative displacements). This relative number will be 1 or −1 if all displacements at a particular BMI bin are in the same direction (either positive or negative). If a particular BMI bin has mixed displacement directions then the relative number will be between −1 and 1. A positive relative number indicates that there are more positive displacements than negative displacements and vice versa (refer to section 3: Results).

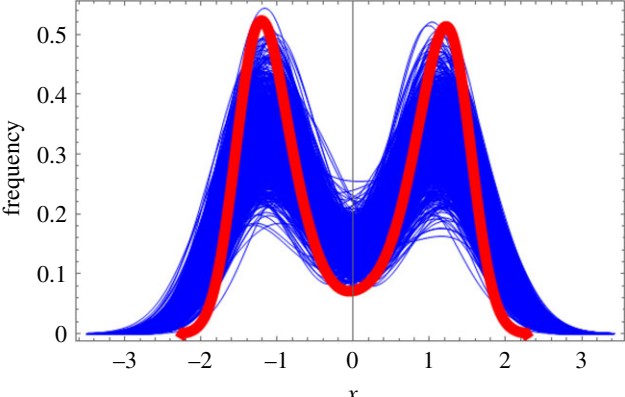

**Figure 6.** The estimated probability densities from the large dataset (in red) consisting of 5000 data-points and the 1000 small datasets (in blue) each consisting of 40 data-points. The datasets were generated from the probability density function, $p(x) = e^{ax^2 - bx^4}/Z$, where $Z$ is the normalizing constant based on the free energy landscape, $F(x) = -ax^2 + bx^4$, from Landau's second order phase transition formalism.

datasets (in blue). We also compare the probability densities estimated from each of the 1000 small datasets against the probability density estimated from the large dataset using the two-sample Kolmogorov–Smirnov test. With a significance level of 0.01, only 7 out of the 1000 small datasets (0.7%) reject the null hypothesis that the large dataset and the small dataset were drawn from the same distribution. From these results, we can say that even with a small dataset, the method is able to generate a close estimate of the true probability density.

## 4. Discussion

Cross-sectional studies are widely prevalent since they require less investment in terms of time, money and effort compared with longitudinal studies. However, since these data lack temporal information, they

cannot be used directly to study the evolution of the underlying dynamics. This temporal information is essential to develop predictive computational models which is the first step towards causal modelling. In this work, we present a method to infer predictive computational models from cross-sectional data using Langevin dynamics. This method can be applied to any system where the data-points are influenced by equal forces and are in (local) equilibrium. That is, at the individual level, there may be continual change in the position of the data-points, but at the population level the distribution of these data-points remains stable. The inferred model will be valid for the time span during which this set of forces remains unchanged. Our proposed method should be viewed as a starting point for inferring temporal dynamics from cross-sectional data. This method only presents a means to obtain an approximate initial estimate of the underlying dynamics from the cross-sectional data without considering any other factors and dependencies. This method is not a causal inference technique in itself. Our model can be made causally interpretable by adding domain expert knowledge to our presented 'baseline' method in the form of (constraints on the) causal relationships between the different variables in the system.

Our method is based on three assumptions. The first assumption is that the distribution depends only on the variable(s) of interest which are chosen to be *dynamic*. The second assumption is that nearby data-points have a statistical tendency to move in similar direction, i.e. downslope of a free energy landscape. Their exact trajectories at a particular time may nevertheless be very different, but this can only be due to the incidental noise which acts on all data-points at all times. The third assumption is that the data-points are sufficiently mixed at the time of observation and are at (local) equilibrium. Thus, we assume that even if there was a major perturbation, a system of data-points has converged to a stable distribution at the time of our observation. That is, at the individual level, there may be continual change in the position of the data-points, but at the population level the distribution of these data-points remains stable. This assumption would be valid for short time spans where we do not expect any major perturbation to the system. The applicability of these assumptions depends crucially on which variable(s) are selected as 'dynamic', which variables are selected as 'confounding' and which variables are selected as 'independent'. This choice of variables as 'dynamic', 'confounding' and 'independent' could be made more accurate with the help of domain expert knowledge.

As opposed to black-box machine learning techniques, our technique is based on interpretable assumptions such that domain expert knowledge can be readily incorporated. That is, the resulting model can be made causally interpretable by adding domain expert knowledge in the form of statements of causal and non-causal relationships. This should lead to increased and more robust predictive power, as is indeed demonstrated by our clustering based on the standard BMI categories.

It is important to realize that the proposed method can only estimate directions of progression, not velocities. This is because, in principle, it is impossible to derive how fast a data-point changes per unit of time from cross-sectional data. In other words, the timescale of the predicted dynamics remains unknown. In some cases, it is possible to estimate a timescale from the data, for instance by quantifying the relative frequency of tipping point transitions in the model and comparing it with knowledge or data about it. It can also be inferred from known statistical properties of the rates of change in reality; for instance, the fact that the maximum sustainable rate of weight loss observed in a population is about 2 kg per month that was used in our previous work [11].

We compare the estimates of the temporal dynamics obtained by our method against two population-based datasets from the public health domain. As these datasets contain data reflective of the weight-related behaviour of a group of individuals, these data are at least in part representative of the outcomes of human interaction and social norms that determine behaviour. Our assumption is that the use of Langevin dynamics can provide an indication of the underlying mechanisms in scenarios where individuals in groups tend to follow norms and adhere to social conventions [23,24]. This is because these 'forces' are hypothesized to lead individuals to move towards the same norm behaviour, making it possible to identify the 'force field' that the individuals are following. There are many cross-sectional studies of human behaviours influenced by social norms, such as physical activity, dietary habits, smoking and alcohol consumption [25–28]. As explained before, cross-sectional data cannot readily be used to develop predictive computational models to study how these behaviours evolve over time. Predictive computational models may, however, be valuable in this context since they enable the assessment of competing hypotheses by allowing us to evaluate hypothetical scenarios *in silico* and simulate the effect of interventions. This is especially advantageous for systems for which comparing counterfactual scenarios would not be possible *in vivo*, as is the case for many systems involving human interaction and social norms. We would, for instance, never be able to conduct an empirical study to assess the effect of group-level social norms versus individual-level weight-related behaviour on body weight [11].

The proposed method can be a useful tool to get an approximate estimate of the underlying dynamics of the system when we only have data for a single time-point. Later, if domain expert knowledge is incorporated, this 'baseline' model could be developed into a causal model and the timescale of the model predictions could be estimated. We believe that the proposed method is sufficiently simple to use as well as interpretable so that it can initiate the iterative development of computational models for any system that can be described as effectively following a free energy landscape and thus help in studying the progression of important processes.

# 5. Conclusion

We have proposed a method to infer predictive computational models from cross-sectional data based on Langevin dynamics for systems where the data-points are influenced by equal forces and are in (local) equilibrium. Our method shows significant predictive power when compared against two population-based longitudinal datasets from the public health domain. The performance of our method could be further improved by taking domain expert knowledge into account. Thus, our method can bootstrap the use of the already abundant cross-sectional datasets to study the evolution of processes and initiate the iterative development of predictive computational models.

Data accessibility. The data that support the findings of this study are available from second parties. Restrictions apply to the availability of these data, which were used under license for this study. Data are available from Lowe *et al.* [16] and Rutters *et al.* [19] on request. The code for the proposed method is written in Mathematica programming language and is available at https://github.com/Pritha17/langevin-crosssectional.

Authors' contributions. R.Q. conceptualized the study. R.Q., P.D. and L.B. performed the mathematical derivations. P.D. carried out the simulations and analysis. P.D. and R.Q. drafted the manuscript with critical input from L.C. and P.M.A.S. All authors approved the final version.

Competing interests. The authors declare no competing interests.

Funding. This work is supported by the NTU Research Scholarship, ZonMw (Netherlands Organization for Health Research and Development, project number: 531003015), Social HealthGames (NWO, the Dutch Science Foundation, project number: 645.003.002), Computational Modelling of Criminal Networks and Value Chains (Nationale Politie, project number: 2454972) and TO_AITION (EU Horizon 2020 programme, call: H2020-SC1-2018-2020, grant number: 848146).

Acknowledgements. We thank Michael Lowe, Prof. (Department of Psychology, Drexel University), for providing us with the College study dataset. We also thank Jeroen Bruggeman, Associate Prof. (Department of Social and Behavioural Sciences, University of Amsterdam), Debraj Roy, Assistant Prof. (Computational Science Lab, University of Amsterdam) and Nadege Merabet, PhD (University of Amsterdam) for reviewing the paper.

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
