## [Peer Review File · Royal Society Open Science]

Review History

RSOS-202147.R0 (Original submission)

Review form: Reviewer 1

Is the manuscript scientifically sound in its present form?

Yes

Are the interpretations and conclusions justified by the results?

No

Is the language acceptable?

Yes

Do you have any ethical concerns with this paper?

No

Have you any concerns about statistical analyses in this paper?

No

Recommendation?

Major revision is needed (please make suggestions in comments)

Comments to the Author(s)

In this paper, the authors propose a method to extract a model from a cross sectional study, thus alleviating the need for longitudinal study that are known to be costly both in time and resource.

Their model rely on the hypothesis that the current system is the result of a diffusion process in a particular energy landscape and that it has reached its equilibrium distribution. From the data, they may thus infer this energy landscape and make predictions based on the diffusion of a point in this landscape.

While this method is well known and validated in physics, I have strong doubts of its validity in the physiological domain. In particular, a strong hypothesis that is not mentioned is that the distribution does not depend (or depend only slightly) on other variables or factors. However, the BMI might be highly correlated with alimentation, activity, guts microbiome, etc. in a way that dominates the diffusion.

Moreover, it is very unclear for me what is the value of the prediction that may be drawn from this kind of model as no time scale, no mechanism and no units may be inferred by the method (a point rightly mentioned by the authors). Typically, what could possibly be the equivalent of page 7 on the BMI example? The presentation of the validation procedure should also be lengthened as I still have difficulties to understand what is exactly covered by the rescaled A . In particular, I would like to see the mathematical description of A^{average} and U_{CI} . Another of my concerns is about the small sample size for some of the data points, while you estimated your procedure with 5000 points, you apply it in the case of 39 points which may give strong under-sampling effects, a word about that should be made in the methods description (typically by evaluating this parameter in the toy model of figure 1).

In figure 3, is it normal that the four panels look exactly the same? And by the way at the end of section 3 you state : « We observe from Fig. 3 that this relative number is between 1 and -1 for most of the BMI bins. » Which is not surprising for a quantity that is rescaled to be between 1 and -1! I guessed that you want to emphasize that this value is usually not near 0 but the formulation is very strange.

As I still have doubts about the method, I would recommend the authors to make their validation procedure more detailed and clear. In particular by giving the mathematical description of all the mentioned quantities.

Decision letter (RSOS-202147.R0)

Dear Dr Quax

The Editors assigned to your paper RSOS-202147 "Inferring temporal dynamics from cross-sectional data using Langevin dynamics" have made a decision based on their reading of the paper and any comments received from reviewers.

Regrettably, in view of the reports received, the manuscript has been rejected in its current form. However, a new manuscript may be submitted which takes into consideration these comments.

We invite you to respond to the comments supplied below and prepare a resubmission of your manuscript. Below the referees' and Editors' comments (where applicable) we provide additional requirements. We provide guidance below to help you prepare your revision.

Please note that resubmitting your manuscript does not guarantee eventual acceptance, and we do not generally allow multiple rounds of revision and resubmission, so we urge you to make every effort to fully address all of the comments at this stage. If deemed necessary by the Editors, your manuscript will be sent back to one or more of the original reviewers for assessment. If the original reviewers are not available, we may invite new reviewers.

Please resubmit your revised manuscript and required files (see below) no later than 11-Oct-2021. Note: the ScholarOne system will 'lock' if resubmission is attempted on or after this deadline. If you do not think you will be able to meet this deadline, please contact the editorial office immediately.

Please note article processing charges apply to papers accepted for publication in Royal Society Open Science (<https://royalsocietypublishing.org/rsos/charges>). Charges will also apply to papers transferred to the journal from other Royal Society Publishing journals, as well as papers submitted as part of our collaboration with the Royal Society of Chemistry (<https://royalsocietypublishing.org/rsos/chemistry>). Fee waivers are available but must be requested when you submit your manuscript (<https://royalsocietypublishing.org/rsos/waivers>).

Thank you for submitting your manuscript to Royal Society Open Science and we look forward to receiving your resubmission. If you have any questions at all, please do not hesitate to get in touch.

on behalf of Dr Francois Fages (Associate Editor) and Marta Kwiatkowska (Subject Editor)
openscience@royalsociety.org

Associate Editor Comments to Author (Dr Francois Fages):

Comments to the Author:

Dear authors

I regret to inform you that your manuscript is rejected due to several weaknesses mentioned in the attached peer-review report. We hope you will find them useful and if you can address them you are welcomed to resubmit a new version of your paper.

Best regards

Reviewer comments to Author:

Reviewer: 1

Comments to the Author(s)

In this paper, the authors propose a method to extract a model from a cross sectional study, thus alleviating the need for longitudinal study that are known to be costly both in time and resource.

Their model rely on the hypothesis that the current system is the result of a diffusion process in a particular energy landscape and that it has reached its equilibrium distribution. From the data,

they may thus infer this energy landscape and make predictions based on the diffusion of a point in this landscape.

While this method is well known and validated in physics, I have strong doubts of its validity in the physiological domain. In particular, a strong hypothesis that is not mentioned is that the distribution does not depend (or depend only slightly) on other variables or factors. However, the BMI might be highly correlated with alimentation, activity, guts microbiome, etc. in a way that dominates the diffusion.

Moreover, it is very unclear for me what is the value of the prediction that may be drawn from this kind of model as no time scale, no mechanism and no units may be inferred by the method (a point rightly mentioned by the authors). Typically, what could possibly be the equivalent of page 7 on the BMI example? The presentation of the validation procedure should also be lengthened as I still have difficulties to understand what is exactly covered by the rescaled A . In particular, I would like to see the mathematical description of A^{average} and U_{CI} . Another of my concerns is about the small sample size for some of the data points, while you estimated your procedure with 5000 points, you apply it in the case of 39 points which may give strong under-sampling effects, a word about that should be made in the methods description (typically by evaluating this parameter in the toy model of figure 1).

In figure 3, is it normal that the four panels look exactly the same? And by the way at the end of section 3 you state : « We observe from Fig. 3 that this relative number is between 1 and -1 for most of the BMI bins. » Which is not surprising for a quantity that is rescaled to be between 1 and -1! I guessed that you want to emphasize that this value is usually not near 0 but the formulation is very strange.

As I still have doubts about the method, I would recommend the authors to make their validation procedure more detailed and clear. In particular by giving the mathematical description of all the mentioned quantities.

===PREPARING YOUR MANUSCRIPT===

===PREPARING YOUR REVISION IN SCHOLARONE===

Author's Response to Decision Letter for (RSOS-202147.R0)

See Appendix A.

RSOS-211374.R0

Review form: Reviewer 1

Is the manuscript scientifically sound in its present form?

Yes

Are the interpretations and conclusions justified by the results?

Yes

Is the language acceptable?

Yes

Do you have any ethical concerns with this paper?

No

Have you any concerns about statistical analyses in this paper?

No

Recommendation?

Accept as is

Comments to the Author(s)

In this paper the authors propose a method to derive temporal information (longitudinal study) from a one point in time measure (cross-sectional study). In order to do so, they make the hypothesis that the system under study is a kind of physical system at equilibrium and derive the approximate "energy" landscape from the distribution of the data-points. They then need some minimal hypothesis to determine a possible field force that may produce this landscape and use these forces to predict the evolution of the initial system.

The authors have done a great work to responds to my previous remarks and in particular to emphasize the hypothesis upon which their method is build and gives the framework in which it

will stay valid. They also discuss in length how this "zero knowledge" model maybe improved and manipulate by field expert to draw information from this kind of result giving a more clear picture of the applicability of the method.

Decision letter (RSOS-211374.R0)

Dear Dr Quax,

I am pleased to inform you that your manuscript entitled "Inferring temporal dynamics from cross-sectional data using Langevin dynamics" is now accepted for publication in Royal Society Open Science.

on behalf of Dr Francois Fages (Associate Editor) and Marta Kwiatkowska (Subject Editor)
openscience@royalsociety.org

Associate Editor Comments to Author (Dr Francois Fages):

Associate Editor

Comments to the Author:

Dear Authors

It is my pleasure to accept as is the resubmission of your paper which correctly took into account the criticisms of the reviewers.

Thank you for your contribution to RSOS.

Reviewer comments to Author:

Reviewer: 1

Comments to the Author(s)

In this paper the authors propose a method to derive temporal information (longitudinal study) from a one point in time measure (cross-sectional study). In order to do so, they make the hypothesis that the system under study is a kind of physical system at equilibrium and derive the approximate "energy" landscape from the distribution of the data-points. They then need some minimal hypothesis to determine a possible field force that may produce this landscape and use these forces to predict the evolution of the initial system.

The authors have done a great work to responds to my previous remarks and in particular to emphasize the hypothesis upon which their method is build and gives the framework in which it will stay valid. They also discuss in length how this "zero knowledge" model maybe improved and manipulate by field expert to draw information from this kind of result giving a more clear picture of the applicability of the method.

Appendix A

Dear Editor,

We appreciate the time and effort that you and the reviewers dedicated in reviewing our manuscript and providing valuable comments. This has substantially improved the quality and clarity of the manuscript. Below we give a comprehensive overview of the issues raised by the reviewer, our response and how we addressed the raised issues. The authors welcome further constructive comments if any.

Yours sincerely, on behalf of the authors,

Dr. Rick Quax

Feedback & Response

We would like to thank the reviewer for the accurate and fair feedback. The reviewer appreciates the novelty of the idea and shows understanding of the goal of the paper.

In the revised manuscript, we have focussed on improving the manuscript by addressing the following main issues based on the reviewer's comments. In summary, first, a detailed account of all the assumptions made in this method have been provided. Second, a detailed discussion of the goal of this method, the systems to which this method is applicable, and the usefulness of this method have been included. Third, mathematical descriptions of all quantities have been provided. In addition, we explain how this method is applicable to the individual level and the population level. We have also included additional theoretical tests and statistical analyses to address the reviewer's concerns.

We have summarized the issues raised by the reviewer in the table below. The authors' remark and the changes introduced in the manuscript are indicated in their respective columns.

Reviewer's remark	Authors' remark	Introduced changes
A strong hypothesis that the distribution does not depend (or depends only slightly) on other variables or factors is not mentioned.	We apologize for this omission. We have now provided a detailed account of all the assumptions in the revised manuscript. We have also better explained how this method applies to the individual level and the population level.	We have added a discussion of the assumptions in paragraphs 4-8 of section 1: Introduction (pages 2-3). We have also summarized the assumptions in paragraph 2 of section 4: Discussion (page 15).
What is the value of the prediction that may be drawn from this kind of model as no time scale, no mechanism and no units may be inferred by the method?	The proposed method can be a useful tool to get an initial estimate of the underlying dynamics of the system when only cross-sectional data is available. Later with the inclusion of domain expert knowledge, the inferred 'baseline' model can be extended into a causal model	The details of systems to which this method is applicable are included in paragraph 9 of section 1: Introduction (pages 3-4). The goal of the method is discussed in paragraphs 10-11 of section 1: Introduction (page 4).

	and the timescale of the model predictions can be estimated.	The usefulness of the method is summarized in the last paragraph of section 4: Discussion (page 16).
What could be the equivalent of page 7 on the BMI example?	In the revised manuscript, we have provided a theoretical test where the free energy landscape is changed by adding a term, which can be considered as analogous to an intervention, and then discussed potential comparison measures between the pre-intervention and post-intervention cases. This theoretical test has been performed for both the two-attractor and single attractor landscapes.	We have added a discussion of the theoretical tests regarding interventions from paragraph 5 to the last paragraph of section 2(b): Numerical algorithm (pages 7-9) and in Figure 2.
Mathematical descriptions of the mentioned quantities in the validation procedure are not provided.	We apologize for the omission. We have now included detailed mathematical descriptions of all quantities used for comparing estimates of the temporal dynamics obtained by our method against longitudinal datasets.	We have added the mathematical expressions of the following quantities in section 2(c): Comparison with longitudinal dataset (pages 9-11):  • P_{PD}^i and P_{ND}^i (equations 2.10 and 2.11) • A^i and $A^{average}$ (equations 2.12 and 2.13) • A_{max}^i and $A_{max}^{average}$ (equations 2.14 and 2.15) • U_{CI} (equation 2.16) We have also included Figure 3 to explain the comparison of the prediction accuracy of our model to random choice.
Performance of the method on a small dataset is not provided.	Thank you for bringing this to our attention. We have improved the prior version by providing performance evaluations of the method on a small dataset.	We have added the comparison of performance results of our model on a large dataset (5000 data-points) and small datasets (40 data-points) in paragraph 5 of section 3: Results (page 14) and Figure 6.
Figure 5 (Figure 3 in previous version) and its explanation in the Results section is not clear.		We have improved the explanation and the purpose of Figure 5 in paragraph 4 of

		section 3: Results (pages 13-14).
--	--	-----------------------------------